# Bisbenzimide compounds inhibit the replication of prototype and pandemic potential poxviruses

Jerzy Samolej,[1] Diogo Correa Mendonca,[2,3] Nicole Upfold,[2,3] Marion McElwee,[2,3] Mariann Landsberger,[1] Artur Yakimovich,[4] Arvind H. Patel,[2,3] Blair L. Strang,[5] Jason Mercer[1]

**ABSTRACT** We previously identified the bisbenzimide Hoechst 33342 (H42) as a potent multi-stage inhibitor of the prototypic poxvirus, the vaccinia virus (VACV), and several parapoxviruses. A recent report showed that novel bisbenzimide compounds similar in structure to H42 could prevent human cytomegalovirus replication. Here, we assessed whether these compounds could also serve as poxvirus inhibitors. Using virological assays, we show that these bisbenzimide compounds inhibit VACV spread, plaque formation, and the production of infectious progeny VACV with relatively low cell toxicity. Further analysis of the VACV lifecycle indicated that the effective bisbenzimide compounds had little impact on VACV early gene expression but inhibited VACV late gene expression and truncated the formation of VACV replication sites. Additionally, we found that bisbenzimide compounds, including H42, can inhibit both monkeypox and a VACV mutant resistant to the widely used anti-poxvirus drug TPOXX (Tecovirimat). Therefore, the tested bisbenzimide compounds were inhibitors of both prototypic and pandemic potential poxviruses and could be developed for use in situations where anti-poxvirus drug resistance may occur. Additionally, these data suggest that bisbenzimide compounds may serve as broad-activity antiviral compounds, targeting diverse DNA viruses such as poxviruses and betaherpesviruses.

**IMPORTANCE** The 2022 mpox (monkeypox) outbreak served as a stark reminder that due to the cessation of smallpox vaccination over 40 years ago, most of the human population remains susceptible to poxvirus infection. With only two antivirals approved for the treatment of smallpox infection in humans, the need for additional anti-poxvirus compounds is evident. Having shown that the bisbenzimide H33342 is a potent inhibitor of poxvirus gene expression and DNA replication, here we extend these findings to include a set of novel bisbenzimide compounds that show anti-viral activity against mpox and a drug-resistant prototype poxvirus mutant. These results suggest that further development of bisbenzimides for the treatment of pandemic potential poxviruses is warranted.

**KEYWORDS** vaccinia virus, antiviral, DNA replication, viral transcription, mpox

The 2022 mpox (monkeypox) outbreak served as a potent reminder of the pandemic potential of poxviruses (1). While existing smallpox vaccines (Imvanex and ACAM2000) provide good protection against mpox infection (2), the cessation of smallpox vaccination has left the global population susceptible to infection by many existing poxviruses for which vaccine efficacy is unknown.

To supplement vaccination, novel drug strategies are required to treat poxvirus infection. In the United States, there are only two drugs approved for human treatment of smallpox: Tembexa (also known as brincidofovir) and TPOXX (also known as Tecovirimat or ST-246) (3–6). In the United Kingdom and European Union, TPOXX is the only

Address correspondence to Blair L. Strang, bstrang@sgul.ac.uk, or Jason Mercer, j.p.mercer@bham.ac.uk.

The authors declare no conflict of interest.

See the funding table on p. 13.

drug approved for the treatment of orthopoxvirus infections including smallpox, mpox, vaccinia, and cowpox. In 2022, TPOXX was approved for the treatment of mpox under the US CDC's expanded access investigational new drug protocol (CDC 2023). Despite its efficacy against many poxviruses, including mpox (7), a single-point mutation within the poxviral genome was sufficient to give rise to TPOXX resistance (4). Therefore, the search for additional anti-poxviral compounds is required.

We have shown that the bisbenzimide Hoechst 33342 (H42) is an effective inhibitor of human and animal poxviruses *in vitro*. H42 was found to inhibit vaccinia virus (VACV) DNA replication and late gene expression (LGE) of the prototype poxvirus, VACV, at low micromolar concentrations (8). H42 is a fluorescent dye that binds within the minor groove of double-stranded DNA, preferentially to adenosine-threonine (AT)-rich regions (9–13). As the anti-poxvirus efficacy of H42 correlated with its membrane permeability and accessibility to the VACV DNA, our data suggested a model in which H42 blocked DNA replication by coating cytoplasmic VACV DNA genomes (8).

A recent report by Falci Finardi and co-workers (14) showed that another bisbenzimide compound, RO-90-7501 [2′-(4-aminophenyl)-[2,5′-bi-1H-benzimidazol]−5-amine] (referred to here as R90) and several of its analogs produced by MRC-Technology (MRT, now LifeArc), could inhibit the replication of human cytomegalovirus (HCMV). This was likely to occur by binding to the HCMV DNA genome and inhibiting the production of HCMV capsids containing genomes. These compounds included MRT00210423, MRT00210424, MRT00210425, MRT00210426, and MRT00210427 (14) (referred to here as M23, M24, M25, M26, and M27). Given the potential broad-spectrum antiviral activity of this extended class of bisbenzimide compounds against viruses with DNA genomes, we set out to determine if R90 or the MRT compounds were effective inhibitors of both prototype and pandemic potential poxvirus replication.

## RESULTS

### MRT compounds M23–M26 display anti-poxvirus activity

R90 and the MRT compounds are structurally similar to H42, with differences in the terminal groups of the compounds and/or substitution of methyl groups for amine groups (Fig. 1A). In previous studies, the compounds R90, M23, M24, and M25, which retained DNA-binding activity inhibited HCMV replication (14). It was noted that M23 was a more effective inhibitor of HCMV replication than R90, which may be due to its greater ability to interact with DNA (14). M26 and M27 were ineffective inhibitors of HCMV replication, likely due to their inability to interact with DNA (14).

To investigate their potential as anti-poxvirus agents, we first tested the ability of R90 and the MRT compounds to block VACV cell-to-cell spread. During infection, poxviruses produce two forms of infectious particles: mature virions (MVs) and extracellular virions (EVs). MVs are more abundant and mediate host-to-host transmission, while EVs contribute to intra-host and cell-to-cell virus spread (15). The VACV replication cycle is efficient, newly assembled EVs are released by 8 hpi, and intracellular replication is complete within 24 h (16). With this in mind, HeLa cells were infected with VACV L-EGFP, a VACV recombinant expressing EGFP from a late viral promoter, at MOI 0.1 to obtain <30% primary infection of the monolayer. To allow sufficient time for virus cell-to-cell spread, infection was allowed to proceed for 24 h in the absence or presence of R90 or the MRT compounds at increasing concentrations. H42 and TPOXX, a tricyclononene carboxamide that inhibits the production of EVs, were included as controls (4). At 24 hpi, samples were analyzed by flow cytometry for the number of EGFP-expressing cells (Fig. 1B, black lines).

As expected, TPOXX and H42 controls prevented virus spread at low to sub-micromolar concentrations. They lowered the percentage of EGFP-expressing cells by 70% and 80%, respectively, at concentrations as low as 0.4 µM. At low micromolar concentrations, R90 showed limited activity, only blocking VACV spread by 20% at 40 µM. M23, M24, and M25 showed concentration-dependent inhibition of VACV spread. While M23 inhibitory activity plateaued between 10 and 40 µM, M24 and M25 completely blocked infection at 40 µM. M26 and M27, which potentially lack DNA-binding activity (14), had limited or no

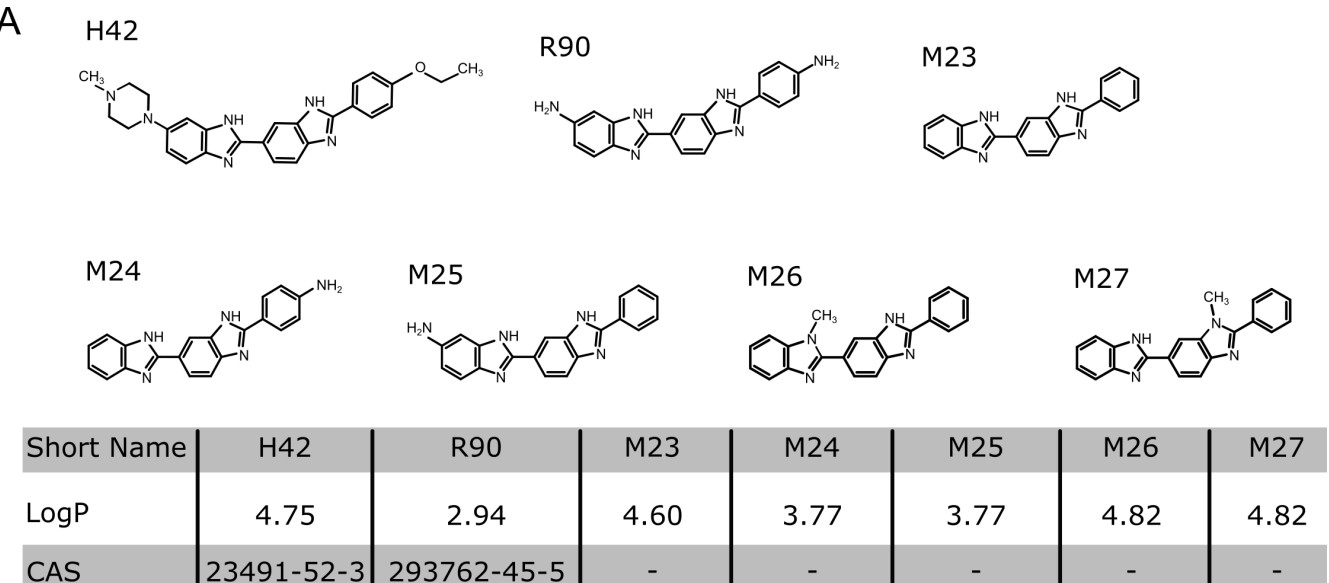

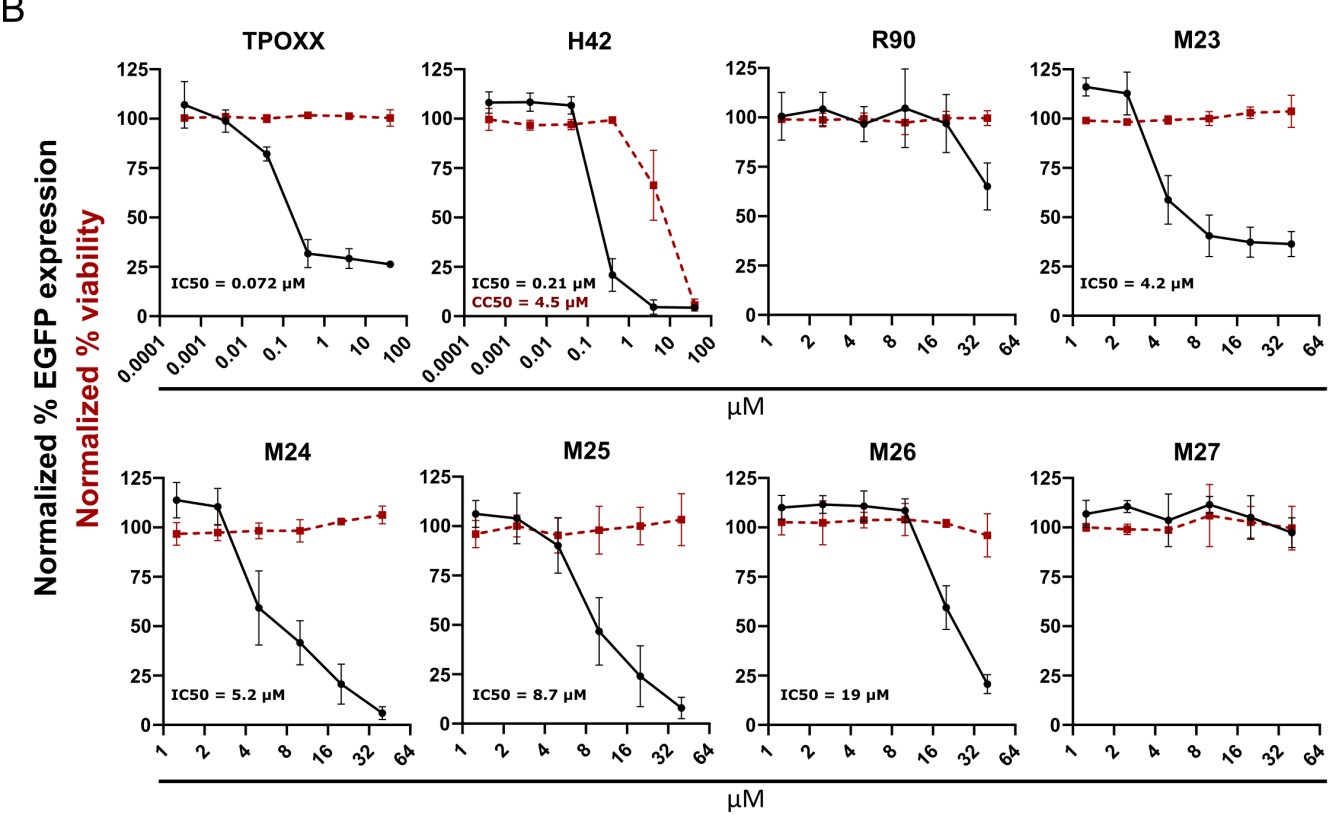

**FIG 1** MRT compounds inhibit the cell-to-cell spread of VACV. (A) Structures of bisbenzimides and MRT compounds H42 (Hoechst 33342), R90 (RO-90-7501), M23 (MRT00210423), M24 (MRT00210424), M25 (MRT00210425), M26 (MRT00210426), and M27 (MRT00210427). Respective partitioning coefficient (LogP) and chemical abstract number (CAS) where applicable are provided below. (B) HeLa cells were infected at an MOI of 0.1 with VACV L-EGFP in the presence of H42, R90, or the MRT compounds. Twenty-four hours post-infection (hpi), cells were quantified by flow cytometry for EGFP and displayed as normalized to infected + DMSO (black lines). Cytotoxicity was assessed using a WST-1 assay and displayed as normalized to DMSO (red dashed lines). Data represent biological triplicates and error bars represent the standard deviations of those data.

obvious anti-VACV activity. M26 displayed modest dose-dependent inhibition of VACV spread at higher concentrations (20–40 µM), while M27 showed no anti-VACV activity.

To ensure the observed anti-viral effects were not due to cellular cytotoxicity caused by the compounds, we measured uninfected cell viability in the presence of each by assaying the ability of cells to metabolize the salt WST-1. Consistent with previous reports, R90 and the MRT compounds had no impact on cell viability at the concentrations used, while H42 displayed toxicity at higher concentrations outside of the effective range of VACV inhibition (Fig. 1B, red lines) (8, 14).

The half-maximal inhibitory concentration (IC50) and half-maximal cell cytotoxicity concentration (CC50) measurements indicate that H42 and the MRT compounds M23-M25 display anti-poxvirus activity, without obvious cellular cytotoxicity (Fig. 1). Compounds R90, M26, and M27 displayed poor or no anti-VACV activity, which may be due to factors that include their poor ability to associate with DNA.

## MRT compounds M23, M24, and M25 reduce VACV yield and block plaque formation

Having determined that R90 and the MRT compounds have varying effects on VACV spread, we next assessed their ability to prevent virus production. The effective concentration against VACV for each compound was determined from the cell-to-cell spread assays shown in Fig. 1. These compound concentrations were used in subsequent experiments, including virus production assays (see figures and figure legends). HeLa cells were infected with wild-type (WT) VACV at an MOI of 1 in the presence of R90, M23, M24, M25, M26, or M27. Infection in the presence of H42 served as a control for the inhibition of virus production. At 24 hpi, cells were harvested, and the infectious virus yield was determined by plaque assay (Fig. 2A). Similar to the results of the cell-to-cell spread assay (Fig. 1), M27 was ineffective and did not impact VACV yield at 24 h, while R90 and M26 showed only a modest (≤1 log) reduction in viral yield. M23, M24, and M25 all reduced virus yield by 3–3.5 logs, confirming the inhibitory effects of these three MRT compounds.

We next assayed plaque formation in the presence of the effective compounds. Focusing on M23, M24, and M25, we infected confluent human retinal pigment epithelial cells (A-RPE-19s) with WT VACV in the presence of the MRT compounds, DMSO, TPOXX, or H42. At 48 hpi, monolayers were stained and assessed for VACV plaque formation (Fig. 2B). M23, M24, and M25 were found to effectively inhibit VACV plaque formation.

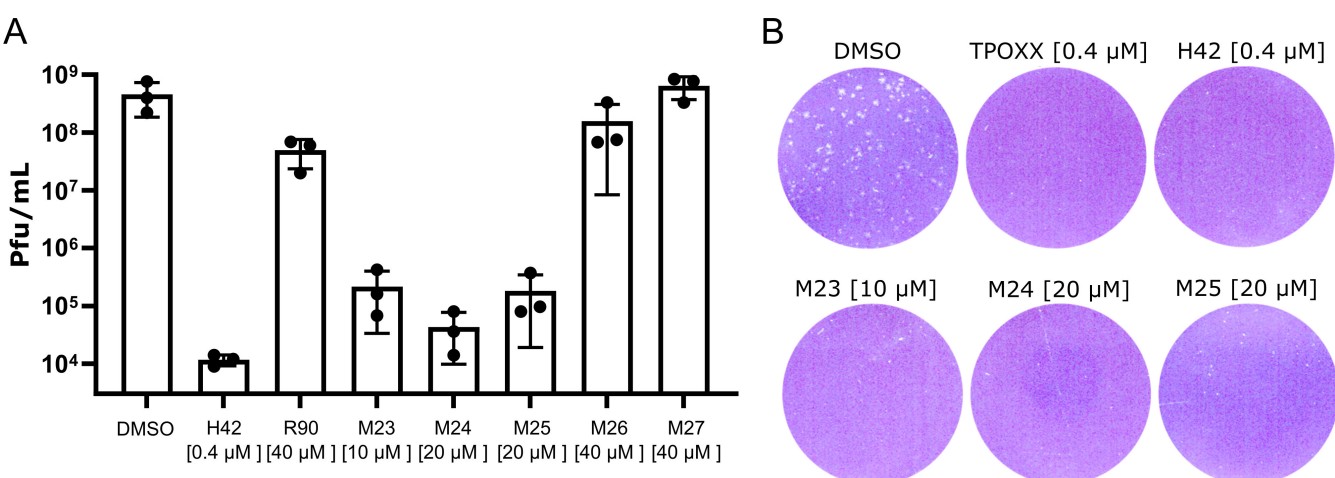

**FIG 2** M23, M24, and M25 reduce virus 24 h yield and inhibit plaque formation. (A) HeLa cells were infected with WT VACV at an MOI of 1 in the presence of the indicated compounds. At 24 hpi, cells were harvested, and VACV progeny was quantified by a titer on BSC40 cells [plaque-forming units (PFU)/mL]. Data represent biological triplicates and error bars represent the standard deviations of those data. (B) A-RPE-19 cells were infected with WT VACV (200 PFU) in the presence of the indicated compounds. At 48 hpi, cells were subjected to fixation and staining to visualize plaques. Experiments were performed in biological duplicates and representative wells of those experiments are shown.

These results indicate that M23, M24, and M25 inhibit VACV production and that they maintain anti-poxvirus activity for at least 48 h. This is consistent with our finding that H42 retained anti-poxvirus activity for at least 72 h (8).

## M23, M24, and M25 inhibit VACV gene expression

We have shown that H42 inhibits poxvirus gene expression (8). To ascertain if the MRT compounds act in a similar fashion, we infected HeLa cells in the presence of M23, M24, or M25 at an MOI of 20 with VACV E-EGFP or VACV L-EGFP (recombinant VACVs expressing EGFP from an early or late viral promoter, respectively). As VACV early gene expression (EGE) occurs before DNA replication and late gene expression after DNA replication, we included cycloheximide (CHX) and cytosine arabinoside (AraC) controls, which inhibit EGE and LGE, respectively. A TPOXX control was also included as a late-stage block that does not affect viral gene expression (8). Cells were harvested at 8 hpi, and the number of E-EGFP- and L-EGFP-expressing cells was quantified by flow cytometry (Fig. 3A). Compared to CHX, H42, M23, M24, and M25 had very modest effects on EGE. Conversely, all bisbenzimide compounds effectively diminished LGE to the levels seen in the presence of AraC. As expected, TPOXX had no effect on either EGE or LGE.

## M23, M24, and M25 reduce the size of VACV replication sites

To corroborate the gene expression results, we infected cells with VACV EGFP-A5 (a VACV recombinant that expresses an EGFP-tagged version of the VACV late core protein A5) in the presence of the various compounds. At 24 hpi, cells were fixed and immunostained for the early VACV protein I3, which is found on uncoated genomes and within VACV replication sites (Fig. 3B). As expected, I3- and A5 co-localized in large replication sites in the presence of either DMSO or TPOXX. In the presence of CHX, which prevents VACV genome uncoating (17), stabilized A5-positive virus cores were seen, but no I3- or A5-positive replication sites were observed. In the presence of AraC, I3-positive uncoated genomes and A5-positive incoming cores were observed, but no VACV replication sites were seen. In the presence of bisbenzimide compounds, small I3-positive, A5-negative replication sites were observed. The replication sites were similar to those seen in the presence of H42, being far smaller and "more compact" than those seen in infected, DMSO-treated controls (8). A5 was not robustly expressed in the presence of any bisbenzimide compound; therefore, no replication sites in which I3 and A5 co-localized were observed. Overall, these data demonstrated that all bisbenzimide compounds inhibit LGE (Fig. 3A), which in turn blocked the development of VACV replication sites (Fig. 3B).

## Bisbenzimides H42, M23, M24, and M25 inhibit mpox infection

We have shown that H42 is effective against orthopox and parapox viruses, suggesting that the bisbenzimides display broad anti-poxviral activity (8). Given the recent worldwide mpox outbreak, we wanted to assess if H42 and MRT compounds could inhibit mpox. Human fetal foreskin fibroblast (Hft) cells were infected with a WT mpox strain, isolated during the recent pandemic, in the presence of H42 or the MRT compounds at various concentrations. At 48 hpi, cells were assessed, in parallel, for cytopathic effect (CPE) and cell cytotoxicity (Fig. 4).

We employed TPOXX as a control for the inhibition of mpox spread (4). As expected, TPOXX displayed potent anti-mpox activity with no apparent toxicity (Fig. 4). H42 and the three MRT compounds were also found to be effective mpox inhibitors with IC50s of 0.075 µM for H42, 4.0 µM for M23, 1.9 µM for M24, and 6.1 µM for M25. H42, M23, and M24 were most effective, completely blocking mpox CPE at 0.22, 13.3, and 4.4 µM, respectively (Fig. 4, blue lines). M25, at its most effective concentration (4.4 µM), reduced CPE to <40%. At these concentrations, H42 caused 50%, M23 48%, M24 27%, and M25 21% cell cytotoxicity, with CC50s of 0.19 µM for H42, 24 µM for M23, 15 µM for M24, and 18 µM for M25. While the Hft cells used for the mpox assay appear to be more sensitive

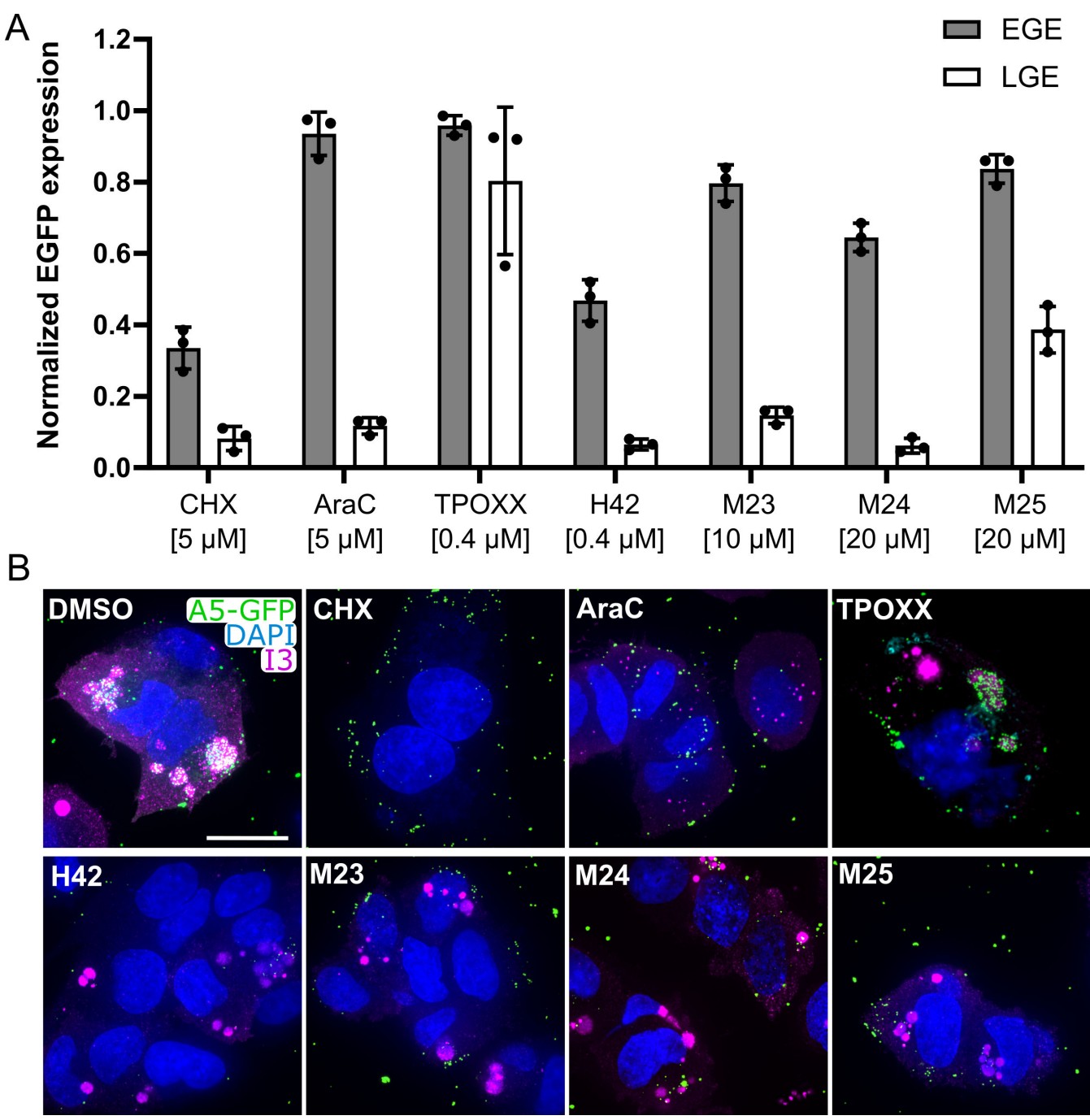

**FIG 3** M23, M24, and M25 block LGE and reduce replication site size. (A) HeLa cells were infected with either VACV E-EGFP or VACV L-EGFP at an MOI of 20 in the presence of the indicated compounds. At 8 hpi, cells were harvested, and EGFP-expressing cells were quantified by flow cytometry. Data displayed as normalized to infected + DMSO = 1. Data represent biological triplicates and error bars represent the standard deviations of those data. (B) HeLa cells were infected with VACV A5-EGFP (green) at an MOI of 20 in the presence of the indicated compounds, concentrations as in panel A. At 24 hpi, fixed cells were immunostained for I3 (magenta), stained with DAPI (blue), and imaged. Scale bar = 20 µm. Experiments were performed in biological duplicates, and representative images of those experiments are shown.

to the compounds than HeLa, A-RPE-19, and BSC cells, overall, these data indicated that bisbenzimides were effective inhibitors of mpox replication.

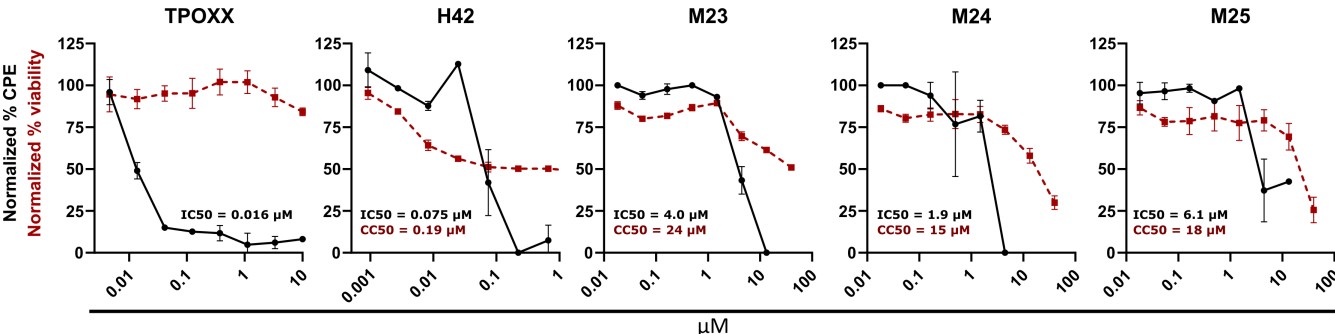

**FIG 4** Inhibitory activity of bisbenzimides and MRT compounds against monkeypox virus. Human fetal foreskin fibroblast cells in 96-well plates were infected with mpox virus (MOI 0.1) in the presence of threefold serial dilutions of TPOXX, H42, M23, M24, or M25. At 48 hpi, virus-induced cytopathic effect (CPE) was quantified from fixed and Coomassie-stained plates. Cell viability was quantified from compound-treated, uninfected cells by measuring the conversion of resazurin to fluorescent resofurin. The percentage of CPE was normalized to infected and uninfected controls, while the percentage of cell viability was normalized to untreated cells and high toxicity control (50% DMSO). For both CPE and toxicity measurements, data represent biological triplicates and error bars represent the standard deviations of those data.

## Bisbenzimides effectively inhibit TPOXX-resistant VACV

TPOXX was approved for the treatment of mpox under the US CDC's expanded access investigational new drug protocol in 2022. It remains the only drug approved for the treatment of both smallpox and mpox. TPOXX targets the viral envelope-wrapping protein F13, which is conserved in all poxviruses. Single point mutations in the gene encoding F13 are known to confer TPOXX resistance to poxviruses *in vitro* and *in vivo* (18, 19).

Thus, we were curious to see if the bisbenzimides could inhibit the spread of a VACV that is partially resistant to TPOXX. For this, we used a VACV expressing the F13 mutant (G277C), which has been described to lower the antiviral efficacy of TPOXX both *in vitro* and *in vivo* (4, 20). To ensure that any phenotypes observed were due to the presence of the G277C mutation and not any other unknown mutations, we generated a control virus— RevG277C—in which the G277C mutant virus was repaired. To examine the effect on virus spread, HeLa cells were infected at a low MOI (0.01) with WT, G277C, or RevG277C. Infections were performed in the presence of DMSO, TPOXX, H42, or M23. Cells were harvested at 24 hpi, and the virus production was quantified by plaque assay (Fig. 5A). TPOXX effectively lowered WT and RevG277C control virus production by >95% compared to the DMSO control. The G277C mutant virus, as expected, showed some resistance to TPOXX (70% reduction). Both H42 and M23 effectively blocked the production of VACV WT, G277C mutant, and RevG277C control viruses, in each case decreasing virus yield by >3.5 log.

To assess the effect of the compounds on plaque formation, A-RPE-19 cells were infected with VACV WT, G277C, or RevG277C in the presence of DMSO, TPOXX (0.4 µM), H42 (0.4 µM), or M23 (10 µM) (Fig. 5B). As expected, in the presence of DMSO, all viruses formed plaques. Both WT and VACV RevG277C control virus were sensitive to TPOXX, while the G277C mutant virus was resistant (albeit, forming somewhat smaller plaques). Consistent with the 24 h yield results (Fig. 5A), H42 and M23 both completely abrogated VACV WT, G277C and RevG277C virus plaque formation.

To confirm that the mechanism of H42 and M23 inhibition remained the same, HeLa cells were infected with G277C or the RevG277C control virus in the presence of DMSO, H42, or M23. At 24 hpi, cells were fixed, and viral replication sites were visualized by immunostaining for I3 (Fig. 5C). In the presence of DMSO, infection with both VACV viruses produced large I3-positive replication sites. In the presence of H42 or M23, I3-positive replication sites were reduced in number and size (Fig. 5C). Collectively, these results indicated that H42 and M23 are effective inhibitors of a TPOXX-resistant mutant VACV.

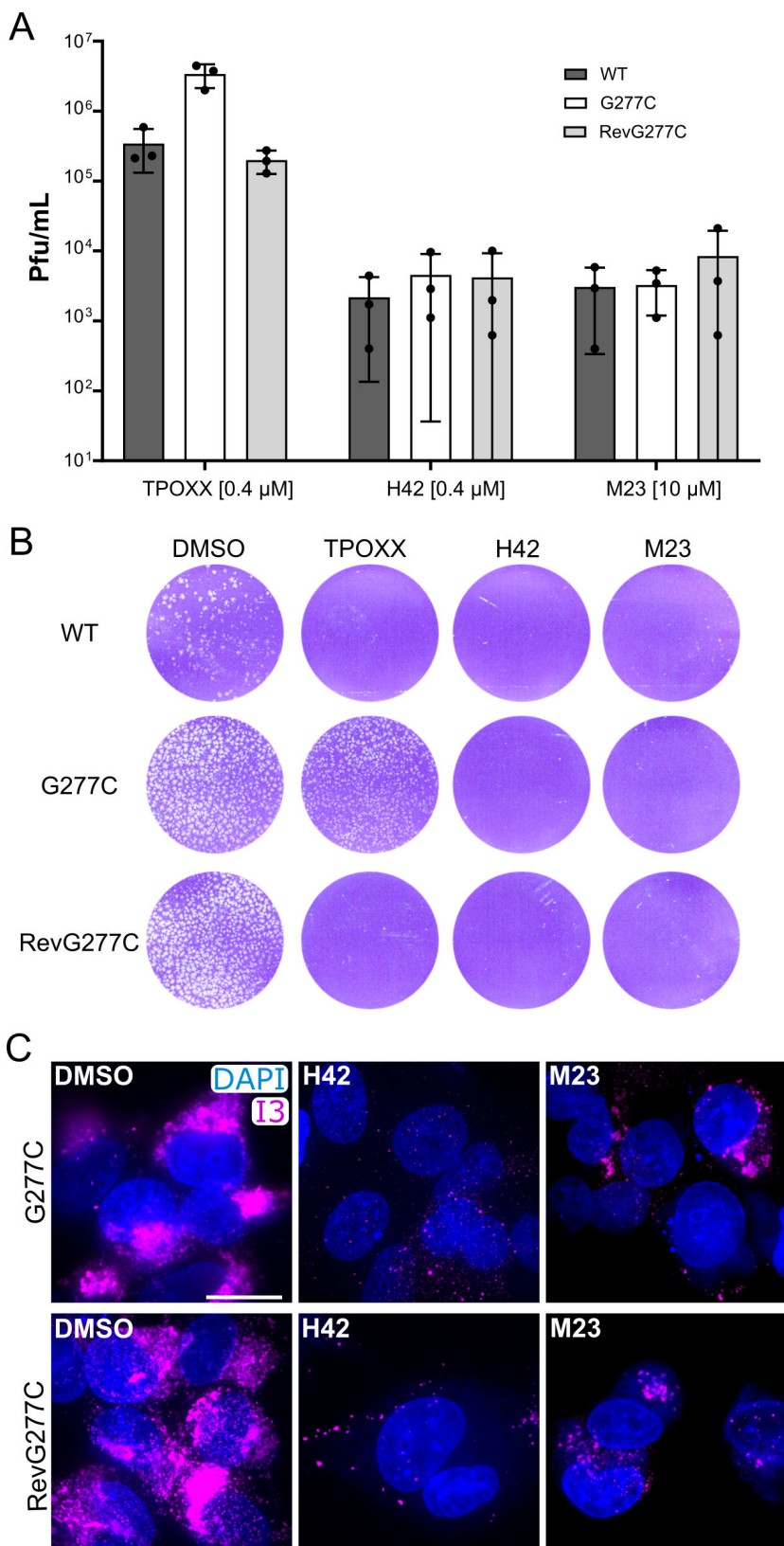

FIG 5 Bisbenzimides are effective against a TPOXX-resistant VACV recombinant. (A) HeLa cells were infected with VACV WT, G277C, or RevG277C viruses at an MOI of 0.01, to measure virus spread, in the presence of the indicated compounds. Cells were harvested at 24 hpi, and the virus yield was determined (Continued on next page)

**FIG 5** (Continued)

by plaque assay. Experiments are biological triplicates and error bars represent the standard deviations of those data. (B) A-RPE-19 cells were infected with 200 plaque-forming units (PFU) of VACV WT, G277C, or RevG277C viruses, in the presence of the indicated compounds at concentrations as in panel A. At 48 hpi, the wells were fixed and stained to visualize virus plaques. Experiments were performed in biological duplicate and representative wells are shown. (C) HeLa cells were infected with G277C or RevG277C virus in the presence of DMSO, H42, or M23 (concentrations like in panel A). At 24 hpi, cells were fixed and immunostained for I3 and stained with DAPI. Experiments were performed in biological duplicate, and representative wells of those experiments are shown. Scale bar = 20 µm.

## DISCUSSION

We have previously shown that bisbenzimides are potent inhibitors of poxvirus infection (8). We found that these compounds, which preferentially bind the minor groove of double-stranded DNA, inhibit infection by blocking DNA replication and post-replicative gene transcription. The bisbenzimide H42 was found to inhibit a range of human and animal poxviruses but was ineffective against several other DNA and RNA viruses, including herpes simplex virus-1 and influenza A.

Here, we tested a series of novel bisbenzimide analogs, with reported activity against HCMV (14), VACV, and mpox. Three of these compounds M23, M24, and M25 proved to be effective inhibitors of prototype (VACV) and pandemic potential (mpox) poxviruses with low cytotoxicity. We show that, like H42 (8), these compounds block poxvirus replication and subsequent LGE.

Poxviruses with varied genomic AT content, ranging from VACV (67% AT) to squirrel pox (33% AT) (21) showed similar sensitivity to H42 (8). Thus, we concluded that the inhibitory efficacy of the bisbenzimides did not correlate with their preferential binding to AT-rich regions of DNA, but with the cytoplasmic accessibility of replicating poxvirus genomes and the lipophilicity of the bisbenzimide compounds, which largely dictate their binding to double-stranded DNA via hydrophobic interactions with adenosine/threonine-rich regions (9, 22, 23). Consistent with this model, in the presence of the MRT compounds, VACV DNA replication sites were small and condensed, and the IC50s of R90, H42, M23, M24, and M25 in HeLa cells largely correlated with their predicted lipophilicity (LogP): H42 > M23 > M24 = M25 > R90 (Fig. 1A). This of course does not preclude other properties of these compounds, which might affect their efficacy, such as membrane permeability and toxicity.

We further show that H42, M23, M24, and M25 were all effective at blocking mpox infection. When assaying mpox replication, TPOXX showed high efficacy and low cytotoxicity compared to the MRT compounds. Despite its *in vitro* potency, as TPOXX targets a viral protein, it is subject to mutational resistance (4, 24). During the 2022 mpox pandemic, TPOXX-resistant mutants were in fact isolated from mpox patients undergoing TPOXX treatment (18, 19). We found that H42 and the MRTs were still effective against a virus that shows resistance to TPOXX. This is not surprising as the bisbenzimides target a different stage in VACV replication than TPOXX (late gene expression and virus assembly/release, respectively) and interact with different factors required for VACV replication (protein F13 and the VACV DNA genome, respectively). As yet, we have been unable to isolate a H42-resistant mutant virus in more than 20 passages of VACV *in vitro* (data not shown) [similar observations have been made by passaging HCMV in the presence of a bisbenzimide (RO)] (14). Thus, the bisbenzimide compounds do not appear to be subject to the development of VACV resistance. This suggests that TPOXX/bisbenzimide co-inhibition studies could be further explored for additive or synergistic effects against poxvirus infection.

It is interesting to compare the mechanisms of action of bisbenzimide compounds on poxviruses and HCMV. At low concentrations, bisbenzimides inhibit VACV gene expression but prevent the formation of HCMV genome-containing capsids (14). While at higher concentrations, both VACV and HCMV genome replication is blocked (8, 14). Given that the anti-viral effects correlate with low concentration bisbenzimide treatment,

it may be worth examining the relationship between HCMV gene expression and genome packaging.

Collectively, this work supports further exploration of bisbenzimides as anti-viral agents. This is supported by the long-standing observations that some bisbenzimide compounds have no obvious adverse effects in mice and have been used with no serious adverse effects in human clinical trials (25). Using the compounds tested here as a platform to generate modified bisbenzimide analogs, in the future, we hope to identify new bisbenzimides with increased potency against poxviruses and perhaps other DNA virus families.

## MATERIALS AND METHODS

### Cells, viruses, and compounds

HeLa (ATCC), BSC-40, A-RPE-19 (kind gift from Frickel lab, UoB), Vero E6, and Primary Human fetal foreskin fibroblasts immortalized by retrovirus transduction to express the catalytic subunit of human telomerase (Hft) (26) were maintained at 37.0°C and 5.0% $CO_2$ in Dulbecco's modified Eagle's medium (DMEM; Gibco, Life Technologies) with the addition of 10% fetal bovine serum (FBS; Sigma), and 1% penicillin-streptomycin (Sigma).

Vaccinia virus strain Western Reserve was used throughout. VACVs used were either wild-type or transgenic, containing EGFP under early VACV gene promoter (VACV E-EGFP), late VACV gene promoter EGFP (VACV L-EGFP), or A5-tagged EGFP inserted into the endogenous A5 locus (VACV EGFP-A5). WT, E-EGFP (27), L-EGFP (27), and EGFP-A5 (28) were previously published. All VACV mature virions were purified from BSC40 cytoplasmic lysates by being pelleted through a 36% sucrose cushion for 90 min at 18,000 × $g$. The virus pellet was resuspended in 1 mM Tris (pH 9.0). The titer [plaque-forming unit (PFU) per milliliter] was determined in BSC40 cells as previously described. Mpox virus (accession number: ON808413; strain designation MPXV CVR-S1) was isolated from a clinical sample in Glasgow in 2022 (29). Vero cells were used to propagate mpox. HFt cells were used in antiviral activity and toxicity assays.

Cycloheximide (Sigma), cytosine arabinoside (Sigma), and TPOXX (generously provided by Dennis Hruby and Douglas Grosenbach, SIGA Technologies, Inc.) were diluted in DMSO and used as in the text, figures, and figure legends. Bisbenzimides Hoechst 33342 (Sigma), RO-90-7501 (Sigma), MRT00210423, MRT00210424, MRT00210425, MRT00210426, and MRT00210427 [all generously provided by Andy Merritt, LifeArc (formerly MRC Technology)] were dissolved in DMSO and used at the concentrations indicated in the text, figures, and figure legends. DMSO was used as a drug carrier control at the same volume as the drug or compound diluted in DMSO.

### Bisbenzimide predicted lipophilicity

To determine the non-ionic consensus partitioning coefficient (LogP) of all structures, we used the AxonChem Marvin cheminformatics suite. Calculations for all structures assumed Cl− and Na+ K + concentrations of 0.1 mol/dm³ each. Tautomerization or resonance were not considered.

### Flow cytometry

HeLa cells in 96-well plates were infected with VACV L-EGFP at an MOI of 0.5 for 24 h (spread assay), or VACV E-EGFP or L-EGFP at an MOI of 20 for 8 h (EGE or LGE assay). After 30 min at room temperature (RT), the inoculant was replaced with DMEM-containing compounds at indicated concentrations. For spread assay: TPOXX, H42, AraC: 40-4-0.4-0.04-0.004-0.0004 µM; R90, M23-27: 40-20-10-5-2.5-1.25 µM. For EGE or LGE, effective concentration with acceptable cytotoxicity derived from the spread assay was used. (Effective concentration was the compound concentration where at least 90% of cells in an infected well were not expressing GFP.) After incubation at 37°C, wells were aspirated and cells detached with trypsin, followed by the addition of 5% BSA

in PBS and fixation with 9% formaldehyde in PBS (for a final 3% FA concentration). The percentage of green fluorescent cells out of all cells was then counted using a Guava easyCyte flow cytometer. Gating was done using "live cells" gate first, and then a "<99% of uninfected cells are below threshold" gate. The results—percentage of cells expressing GFP—were then normalized to infected, DMSO-treated controls (DMSO = 1). IC50/CC50 concentrations were determined in GraphPad Prism using a four-parameter logistic nonlinear regression model (inhibitor) vs response (four parameters).

## Cytotoxicity

Cytotoxicity was assessed using Abcam's Quick Cell Proliferation Assay Kit II (WST-1) following the manufacturer's instructions. Briefly, HeLa cells in 96-well plates were incubated for 24 h at 37°C in the compound concentrations mirroring those concentrations used in the flow cytometry spread assay. WST solution was then added to each well and incubated for 3 h, followed by absorbance measurement at 460 nm, corrected by subtracting absorbance in wells without cells but with media. Values were then normalized to cells incubated without any compounds.

## Virus yield and spread assays

HeLa cell monolayers in 6-well plates were infected with VACV WT at an MOI of 1 (24 h yield) or MOI of 0.01 (24 h spread) in the presence of the specified compound. At 24 hpi, cells were collected and centrifuged, and the pellet was resuspended in 100 µL of 1 mM Tris (pH 9.0). Cells were then freeze-thawed three times to lyse the cells, and the lysate solution was serially diluted to determine the PFU per milliliter by plaque assay on BSC40 cell monolayers.

## Plaque inhibition assays

A-RPE-19 cells grown in 12 wells were infected with 200 PFU of VACV WT, G277C, or G277C-rev in the presence of the specified compound at 37°C. Forty-eight hours post-infection, cells were fixed and stained with 0.1% crystal violet in 4% formaldehyde. Plate images were digitally captured using a desktop scanner (Canon).

## Immunofluorescence microscopy

HeLa were cells seeded on CellView slides (Greiner Bio-One). They were infected with VACV EGFP-A5 for 30 min at RT. The inoculant was then replaced with the indicated compounds in the text, figures, and figure legends. After 20 h at 37°C, cells were washed and fixed with 4% EM grade FA in PBS. They were permeabilized and blocked simultaneously in 0.5% Triton-X 1000 in 5% BSA in PBS. Anti-I3 antibody (generously provided by Jakomine Krijnse Locker, Institute Pasteur) was used at 1:1,000. All secondary antibodies (goat anti-mouse-AF488 and goat anti-rabbit-AF647, Invitrogen) were used at 1:1,000. Primary I3 antibody was added for 60 min at RT, followed by a wash and 60 min RT staining with secondary antibody and DAPI. Images were captured using a 100× oil immersion objective (NA 1.45) on a VT-iSIM microscope (Visitech, Nikon Eclipse TI), using 488 and 640 nm laser frequencies for excitation.

## Mpox antiviral activity and toxicity assays

HFt cells were seeded in 96-well plates (Costar) at a density of $1 \times 10^4$ cells per well and incubated for 24 h. Three hours prior to infection, the cells were incubated with threefold serial dilutions of each compound prepared in the infection medium (DMEM containing 2% FBS). For mpox antiviral assays, the plates were transferred to a CL3 facility before each well was infected with an equal volume of infection medium containing mpox virus at an MOI of 0.1 ($1.4 \times 10^3$ PFU) per well. Following incubation for 48 h, cells were fixed in 8% formaldehyde in PBS and stained with Coomassie blue. The dried plates were scanned using a Pherastar SFX plate reader (BMG) at an optical density of 595 nm to

quantify the level of cytopathic effect. For toxicity assays, an equal volume of infection medium without virus was added to each well. Following 48 h, 10 µL of resazurin (Sigma R7017) prepared at a concentration of 0.5 mM in PBS was added to each well. After a 2 h incubation period, resofurin was quantified by measuring fluorescence intensity (Ex530/Em560) using a Varioskan LUX microplate reader (Thermo Scientific). The percentage of virus replication was calculated by normalizing well clearance to infected and uninfected DMSO controls, while the percentage of cell viability was determined by normalizing values to untreated cells and our high toxicity control (50% DMSO). IC50/CC50 concentrations were determined in GraphPad Prism using a four-parameter logistic nonlinear regression model (inhibitor) vs normalized response (four parameters).

## ACKNOWLEDGMENTS

We thank Andy Merritt (LifeArc) for the generous provision of reagents and his support of the project throughout. We thank Dennis Hruby and Douglas Grosenbach (SIGA Technologies, Inc.) for providing TPOXX.

B.L.S. was supported by St George's, University of London. J.M. was supported by the Medical Research Council (MC_PC_19029) and the BBSRC-funded mpox rapid response grant BB/X011607/1.

The mpox virus work was carried out by CRUSH: Antiviral Drug Screening and Resistance Hub at the MRC-University of Glasgow Centre for Virus Research (CVR).

Funding for this research was supported by the BBSRC-funded mpox rapid response grant BB/X011607/1, LifeArc COVID-19 award, and the MRC core award MC_UU_00034/9 (A.H.P.). This work was partially funded by the Center for Advanced Systems Understanding (CASUS), which is financed by Germany's Federal Ministry of Education and Research (BMBF) and by the Saxon Ministry for Science, Culture, and Tourism (SMWK) with tax funds on the basis of the budget approved by the Saxon State Parliament (A.Y.).

J.S. analyzed and curated the data, designed the methodology, supervised the study, wrote the original draft, and reviewed and edited the manuscript. D.C.M. and N.U. conceptualized the study, designed the methodology, performed the investigation, analyzed and curated the data, and reviewed and edited the manuscript. M.Mc.E. and M.L. performed the investigation. A.Y. performed the investigation and reviewed and edited the manuscript. A.H.P. supervised the study, reviewed and edited the manuscript, and acquired funding. B.L.S. conceptualized the study, designed the methodology, wrote the original draft, and reviewed and edited the manuscript. J.P.M. conceptualized the study, designed the methodology, wrote the original draft, reviewed and edited the manuscript, supervised the study, administrated the project, and acquired funding.

## AUTHOR AFFILIATIONS

[1]Insititute of Microbiology and Infection, University of Birmingham, Birmingham, United Kingdom
[2]MRC-University of Glasgow Centre for Virus Research, Glasgow, United Kingdom
[3]CVR-CRUSH, MRC-University of Glasgow Centre for Virus Research, Glasgow, United Kingdom
[4]Center for Advanced Systems Understanding (CASUS), Helmholtz-Zentrum Dresden-Rossendorf e.V. (HZDR), Görlitz, Germany
[5]Institute for Infection and Immunity, St George's, University of London, London, United Kingdom

## AUTHOR ORCIDs

Jerzy Samolej  http://orcid.org/0000-0003-0615-3297
Mariann Landsberger  http://orcid.org/0000-0003-0863-7172
Artur Yakimovich  http://orcid.org/0000-0003-2458-4904
Arvind H. Patel  http://orcid.org/0000-0003-4600-2047

Blair L. Strang  http://orcid.org/0000-0001-9407-1974
Jason Mercer  http://orcid.org/0000-0003-1466-9541

## FUNDING

| Funder | Grant(s) | Author(s) |
|---|---|---|
| UKRI | Medical Research Council (MRC) | MC_PC_19029, MC_UU_00034/9 | Jason Mercer |
| | | Diogo Correa Mendonca |
| | | Nicole Upfold |
| | | Marion McElwee |
| | | Arvind H. Patel |
| UKRI | Biotechnology and Biological Sciences Research Council (BBSRC) | BB/X011607/1 | Jason Mercer |
| | | Jerzy Samolej |
| | | Diogo Correa Mendonca |
| | | Nicole Upfold |
| | | Marion McElwee |
| | | Arvind H. Patel |

## AUTHOR CONTRIBUTIONS

Jerzy Samolej, Data curation, Formal analysis, Methodology, Supervision, Writing – original draft, Writing – review and editing | Diogo Correa Mendonca, Conceptualization, Data curation, Formal analysis, Investigation, Methodology, Writing – review and editing | Nicole Upfold, Conceptualization, Data curation, Formal analysis, Investigation, Methodology, Writing – review and editing | Marion McElwee, Investigation | Mariann Landsberger, Investigation | Artur Yakimovich, Investigation, Writing – review and editing | Arvind H. Patel, Funding acquisition, Supervision, Writing – review and editing | Blair L. Strang, Conceptualization, Methodology, Writing – original draft, Writing – review and editing | Jason Mercer, Conceptualization, Funding acquisition, Methodology, Project administration, Supervision, Writing – original draft, Writing – review and editing

## ADDITIONAL FILES

The following material is available online.

Open Peer Review

**PEER REVIEW HISTORY (review-history.pdf).** An accounting of the reviewer comments and feedback.

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
