## [Reviewer comments · Microbiology Spectrum]

Microbiology Spectrum

Bisbenzimidazole compounds inhibit replication of prototype and pandemic potential poxviruses

Jason Mercer, Jerzy Samolej, Diogo Mendonca, Nicole Upfold, Marion McElwee, Mariann Landsberger, Artur Yakimovich, Arvind Patel, and Blair Strang

Corresponding Author(s): Jason Mercer, University of Birmingham

Review Timeline:

Submission Date:	December 4, 2023
Editorial Decision:	January 4, 2024
Revision Received:	January 15, 2024
Accepted:	February 1, 2024

Editor: Luciana Costa

Reviewer(s): The reviewers have opted to remain anonymous.

Transaction Report:

DOI: <https://doi.org/10.1128/spectrum.04072-23>

Re: Spectrum04072-23 (Bisbenzimidazole compounds inhibit replication of prototype and pandemic potential poxviruses)

Dear Prof. Jason Mercer:

Thank you for the privilege of reviewing your work. Below you will find my comments, instructions from the Spectrum editorial office, and the reviewer comments.

All reviewers' comments have to be strictly followed, especially reviewer #1. A new experiment was requested, which is of great importance, authors should consider providing the new data.

Revision Guidelines

Sincerely,
Luciana Costa
Editor
Microbiology Spectrum

Reviewer #1 (Comments for the Author):

Major comments:

Line 86: The authors mention the use of the compounds to block cell-cell spread of the virus, but do not elaborate on how this

mechanism of infection differs from conventional viral infection. Is this a particle independent mechanism?

Fig 2A: The data shown here only has a single time point of 24h. It is important to understand if virus production remains diminished at later time points in the presence of the compounds. Growth curves for upto 72h would be informative.

Fig 3: Please justify the use of a higher MOI of 20 in this assay as compared to the MOI of 1 in the previous assay.

Fig 3B: There are green punctae present in the CHX treated cells- are these incoming viruses? If so, the description in the text should be modified to indicate as such.

Line 153: I think these should say A5 positive incoming genomes, but no I3 positive replication sites.

Fig 5: What is the difference between the wt virus and the RevG277C virus? It should be explained how they differ to understand their use in this assay.

Fig 5A: Please show plaque titers in a log scale on the Y axis. The error bars in the H42 and M23 treated samples are very widely distributed- please comment.

Line 205: Indicate the figure number here- Also, there is no I3 staining visible in the H42 treated cells infected with the G277C virus. There should be some staining since replication is not completely abrogated at 24h as seen in Fig 5A.

Minor comments:

Line 36: Mention full name before introducing abbreviations for the virus. Also, there is a typo in the sentence- it should read "...cessation of small pox.."

The manuscript can be polished to make reading more fluid.

Reviewer #2 (Comments for the Author):

Samolej et al describe the in vitro activity of bisbenzimidazole compounds and their antiviral activity on vaccinia virus. The paper is well written and informative. My only critique is that no statistical methods were applied to the data. Perhaps this is because the data was normalized to the control in all experiments. Still, it would be interesting to see the statistical comparison of each compound relative to the control and the other compounds.

Response to reviewers: Spectrum04072-23; Bisbenzimidazole compounds inhibit replication of prototype and pandemic potential poxviruses.

We thank the reviewers for their helpful comments and suggestions.

Reviewer #1 (Comments for the Author):

Major comments:

Line 86: The authors mention the use of the compounds to block cell-cell spread of the virus, but do not elaborate on how this mechanism of infection differs from conventional viral infection. Is this a particle independent mechanism?

We have now added additional information to the manuscript regarding VACV infectious virion and replication kinetics. Poxviruses make two types of infectious particle during infection: mature virions (MVs) and extracellular virions (EVs). EVs are first released at 8 hours post infection (hpi) and mediate virus spread. VACV intracellular replication is complete by 24 hpi.

Fig 2A: The data shown here only has a single time point of 24h. It is important to understand if virus production remains diminished at later time points in the presence of the compounds. Growth curves for up to 72h would be informative.

As above, the VACV replication cycle is ~8 hours and is complete/plateaus by 24 hpi. As such the “24-hour yield” has been the field standard for measuring infectious virion production for over 70 years. (Furness & Youngner 1959)

Given that the bisbenzimidazoles block the early stages of infection and prevent virus spread, assaying production beyond 24 h will not provide any additional information.

However, Figure 2B shows representative 48 h plaque formation in the presence of TPOXX, H42, M23, M24, and M25. We observed no sign of plaque formation in the presence of these compounds at 48 hpi. In addition, we have also assessed the impact of H42 on VACV plaque formation at 72 hpi and observed no break-through plaque formation at this extended time point (Yakimovich *et al* 2017; Figure 4A,B).

Collectively this data indicates that virus production remains diminished at later time points. We have alluded to this extended timeframe of anti-poxvirus activity and added reference to our previous findings with H42 .

Fig 3: Please justify the use of a higher MOI of 20 in this assay as compared to the MOI of 1 in the previous assay.

As this was a gene expression experiment, we use higher MOIs to both synchronize infection and ensure that all cells are infected. As a standard in the field, 24 h yield experiments are performed at an MOI of 1.

Fig 3B: There are green punctae present in the CHX treated cells- are these incoming viruses? If so, the description in the text should be modified to indicate as such.

We have modified the description in the text to read, “In the presence of CHX, which prevents VACV genome uncoating (Mercer *et al* 2012), stabilized A5-positive virus cores, but no I3- or A5-positive replication sites were observed”

Line 153: I think these should say A5 positive incoming genomes, but no I3 positive replication sites.

I3-positive incoming genomes is correct. As indicated in the manuscript, the early VACV protein I3 is found on uncoated genomes and within VACV factories. The magenta dots in the AraC panel are uncoated I3-positive VACV genomes.

For clarity, we have expanded the description of the findings to read, “In the presence of AraC, I3-positive uncoated genomes and A5-positive incoming cores were observed, but no VACV replication sites were seen”.

Fig 5: What is the difference between the wt virus and the RevG277C virus? It should be explained how they differ to understand their use in this assay.

The RevG277C virus is no different from WT. We repaired the mutation in the virus to assure that the G277C virus does not contain any other mutation(s) that may confer the observed phenotypes.

This information has been added to the manuscript.

Fig 5A: Please show plaque titers in a log scale on the Y axis.

We have changed the y-axes in Figures 2A and 5A to log scale.

The error bars in the H42 and M23 treated samples are very widely distributed-please comment.

At such low numbers of plaques in the 1×10^{-3} - 1×10^{-2} range we tend to see larger variation in the titers between biological replicates. Despite this, we find that the bisbenzimidazole are still very effective (3 log) inhibitors of TPOXX-resistant VACV.

Line 205: Indicate the figure number here-

Thank you for pointing this out. We have added the figure number to the text.

Also, there is no I3 staining visible in the H42 treated cells infected with the G277C virus. There should be some staining since replication is not completely abrogated at 24h as seen in Fig 5A.

We thank the reviewer for pointing this out. As we previously reported (Yakimovich *et al* 2017) and can be seen in Figure 3A of this manuscript, H42 does impact VACV EGE. We routinely see cell-to-cell variability in I3 staining when using these compounds.

For consistency, we have swapped the image with another field from the same experiment which displays I3 staining more similar to that seen in the RevG277C sample.

Minor comments:

Line 36: Mention full name before introducing abbreviations for the virus. Also, there is a typo in the sentence- it should read "... cessation of small pox.."

We have corrected both instances.

The manuscript can be polished to make reading more fluid.

We have edited the manuscript for readability.

Reviewer #2 (Comments for the Author):

Samolej et al describe the in vitro activity of bisbenzimidazole compounds and their antiviral activity on vaccinia virus. The paper is well written and informative. My only critique is that no statistical methods were applied to the data. Perhaps this is because the data was normalized to the control in all experiments. Still, it would be interesting to see the statistical comparison of each compound relative to the control and the other compounds.

As pointed out by the reviewer, we did not include statistics as all experiments were normalized to control samples. In addition, the data is rather binary in the sense that we either see no effect or obvious and substantial effects with the various compounds.

References:

Furness G, Youngner JS, One-step growth curves for vaccinia virus in cultures of monkey kidney cells, *Virology*, 1959, Volume 9, Issue 3. [https://doi.org/10.1016/0042-6822\(59\)90130-8](https://doi.org/10.1016/0042-6822(59)90130-8).

Yakimovich A, Huttunen M, Zehnder B, Coulter LJ, Gould V, Schneider C, Kopf M, McInnes CJ, Greber UF, Mercer J. Inhibition of Poxvirus Gene Expression and Genome Replication by Bisbenzimidazole Derivatives. *J Virol*. 2017 Aug 24;91(18):e00838-17. doi: 10.1128/JVI.00838-17. PMID: 28659488; PMCID: PMC5571260.

Mercer J, Snijder B, Sacher R, Burkard C, Bleck CK, Stahlberg H, Pelkmans L, Helenius A. RNAi screening reveals proteasome- and Cullin3-dependent stages in vaccinia virus infection. *Cell Rep*. 2012 Oct 25;2(4):1036-47. doi: 10.1016/j.celrep.2012.09.003. PMID: 23084750.

Re: Spectrum04072-23R1 (Bisbenzimidazole compounds inhibit replication of prototype and pandemic potential poxviruses)

Dear Prof. Jason Mercer:

Your manuscript has been accepted, and I am forwarding it to the ASM production staff for publication. Your paper will first be checked to make sure all elements meet the technical requirements. ASM staff will contact you if anything needs to be revised before copyediting and production can begin. Otherwise, you will be notified when your proofs are ready to be viewed.

Sincerely,
Luciana Costa
Editor
Microbiology Spectrum

Reviewer #1 (Comments for the Author):

The authors have responded sufficiently to the previous comments.

Reviewer #2 (Comments for the Author):

The manuscript is acceptable for publication.